# Dynamic Changes in the Endocannabinoid System during the Aging Process: Focus on the Middle-Age Crisis

**DOI:** 10.3390/ijms231810254

**Published:** 2022-09-06

**Authors:** Prakash Nidadavolu, Andras Bilkei-Gorzo, Felix Effah, Este Leidmaa, Britta Schürmann, Moritz Berger, Laura Bindila, Matthias Schmid, Beat Lutz, Andreas Zimmer, Alexis Bailey

**Affiliations:** 1Institute of Molecular Psychiatry, Medical Faculty, University of Bonn, 53125 Bonn, Germany; 2Pharmacology Section, Institute of Medical and Biomedical Education, St George’s University of London, London SW17 0RE, UK; 3Department of Medical Biometry, Informatics and Epidemiology, Faculty of Medicine, University of Bonn, 53125 Bonn, Germany; 4Institute of Physiological Chemistry, University Medical Center of the Johannes Gutenberg University Mainz, 55131 Mainz, Germany; 5Leibniz Institute for Resilience Research (LIR), 55122 Mainz, Germany

**Keywords:** aging, endocannabinoid system, cognitive functions

## Abstract

Endocannabinoid (eCB) signaling is markedly decreased in the hippocampus (Hip) of aged mice, and the genetic deletion of the cannabinoid receptor type 1 (CB1) leads to an early onset of cognitive decline and age-related histological changes in the brain. Thus, it is hypothesized that cognitive aging is modulated by eCB signaling through CB1. In the present study, we detailed the changes in the eCB system during the aging process using different complementary techniques in mouse brains of five different age groups, ranging from adolescence to old age. Our findings indicate that the eCB system is most strongly affected in middle-aged mice (between 9 and 12 months of age) in a brain region-specific manner. We show that 2-arachidonoylglycerol (2-AG) was prominently decreased in the Hip and moderately in caudate putamen (CPu), whereas anandamide (AEA) was decreased in both CPu and medial prefrontal cortex along with cingulate cortex (mPFC+Cg), starting from 6 months until 12 months. Consistent with the changes in 2-AG, the 2-AG synthesizing enzyme diacylglycerol lipase α (DAGLα) was also prominently decreased across the sub-regions of the Hip. Interestingly, we found a transient increase in CB1 immunoreactivity across the sub-regions of the Hip at 9 months, a plausible compensation for reduced 2-AG, which ultimately decreased strongly at 12 months. Furthermore, quantitative autoradiography of CB1 revealed that [^3^H]CP55940 binding markedly increased in the Hip at 9 months. However, unlike the protein levels, CB1 binding density did not drop strongly at 12 months and at old age. Furthermore, [^3^H]CP55940 binding was significantly increased in the lateral entorhinal cortex (LEnt), starting from the middle age until the old age. Altogether, our findings clearly indicate a middle-age crisis in the eCB system, which could be a potential time window for therapeutic interventions to abrogate the course of cognitive aging.

## 1. Introduction

The endocannabinoid (eCB) system is a neuromodulatory system that engages a vast variety of physiological processes, including nociception, appetite, mood regulation, cognitive functions, and neurogenesis. It comprises the cannabinoid receptors type 1 and 2 (CB1 and CB2), the cognate endogenous eCB ligands anandamide (AEA) and 2-arachidonoylglycerol (2-AG), and the enzymes that achieve their biosynthesis and degradation. While *N*-acylphosphotidylethanolamine-specific phospholipase D (NAPE-PLD) and calcium-dependent *N*-acyltransferase (NAT) are mainly responsible for AEA synthesis, α/β hydrolase domain-containing protein 4 (ABHD4) and fatty acid amide hydrolase (FAAH) contributes to its degradation. The 2-AG is mostly synthesized by diacylglycerol lipase α (DAGLα) and degraded by monoacylglycerol lipase (MAGL) [1,2,3]. The presence of an active eCB system was detected as early as gestational day 14 in rat brains and was reported to undergo dynamic changes until the early postnatal development [4]. Changes in eCB signaling during adolescence contribute to the maturation of corticolimbic circuitry by creating a balance between excitatory and inhibitory neurotransmission [5].

Several studies suggest that CB1 signaling has a central role in the aging process. For example, genetic deletion of CB1 accelerates cognitive aging (early onset of learning and memory impairments) accompanied by neuronal loss and inflammation in the hippocampus (Hip) [6,7,8]. Further, naturally aged mice exhibit a marked decrease in CB1 signaling, specifically in the Hip, as shown by a prominent reduction in 2-AG and DAGLα protein levels [9]. Decreased eCB signaling is one of the many factors that cause learning and memory impairments associated with the natural aging process. Enhancing CB1 signaling in 12- and 18-month-old mice with a chronic low dose THC (3 or 1 mg/kg/day) completely restored learning and memory performances to the level of 2-month-old mice [10,11]. These studies suggest that THC-induced CB1 signaling has the potential to alter the course of cognitive aging. Interestingly, the therapeutic effects of THC are clearly age-dependent, since the same 3 mg/kg/day dose of THC resulted in adverse effects (cognitive dysfunction) in young individuals corresponding to clinical reports from human studies [10,12,13,14,15]. Moreover, repeated stimulation of CB1 with higher doses of THC (10 mg/kg) leads to receptor desensitization and tolerance development in young mice, whereas aged mice develop no such effects [16]. A difference in the intensity of eCB signaling in young and old mice could be a plausible explanation for the differential effects of THC in the respective age groups.

As stated above, there is evidence showing that 2-AG and DAGLα protein levels are markedly reduced in the Hip of 12–15-month-old mice [9]. Additionally, the basal G-protein coupling of CB1 is also prominently decreased in both the hippocampus and limbic forebrain of aged mice [16,17]. However, these studies only showed the changes in CB1 signaling in old age, but how the eCB system is altered during the aging process is still an open question. Determining a time window at which the eCB system is most affected during the aging process was, thus, an important goal of the present study. We used five different age groups (2-, 6-, 9-, 12-, and 18-month-old mice), ranging from adolescence to old age. We chose four different brain regions (Hip; hypothalamus (Hy); caudate putamen (CPu); medial prefrontal cortex along with cingulate cortex (mPFC+Cg)) to analyze the eCB signaling which were known to be affected in aged mice [9,16,17]. Our findings show undulating changes in the eCB system during the aging process, with major changes specifically detected during middle age. Our study uncovers a middle-age crisis in the eCB system, a potential time window that can be further explored for therapeutic interventions.

## 2. Results

### 2.1. Levels of 2-Archidonoylglycerol and Anandamide Fluctuate in an Age- and Region-Specific Manner

To determine age-related changes in the eCB tone, we first analyzed the two major eCBs, 2-AG and AEA, in five different age groups. The 2-AG levels varied in different brain regions, as demonstrated by a significant effect of “brain region” (F_(3, 107)_ = 199.83; *p* < 0.0001). Furthermore, the age factor also significantly influenced the 2-AG levels (F_(4, 45)_ = 6.89; *p* = 0.0002). However, no significant interaction was observed between the two factors, age and brain region (F_(12, 107)_ = 1.44; *p* = 0.1603). Pairwise Dunnett’s multiple comparison tests revealed significant changes only in the Hip and CPu across the brain regions tested. A prominent decrease in 2-AG levels was observed as early as 9 months in both the Hip and CPu compared to the youngest group tested. Although moderately, the decrease in 2-AG levels persisted until 12 months in the Hip, whereas no such decrease was found in the CPu beyond 9 months of age (Figure 1A).

Similarly, AEA levels were also altered significantly in a brain region-specific (F_(3, 107)_ = 257.95; *p* < 0.0001) and an age-dependent manner (F_(4, 45)_ = 4.72; *p* = 0.0029) with a significant interaction (F_(12, 107)_ = 4.10; *p* < 0.0001) between the two main factors. Pairwise Dunnett’s multiple comparisons showed significant changes only in mPFC+Cg and CPu, but not in Hy and Hip. Among the age groups, a significant decrease in AEA levels was observed as early as 6 months. These became more prominent at 9 months and persisted until 12 months of age in both mPFC+Cg and CPu (Figure 1B). Moreover, arachidonic acid (AA), immediate precursor and the degradation product of 2-AG and AEA, was also altered significantly in a brain region-specific (F_(3, 107)_ = 322.34; *p* < 0.0001) and an age-dependent manner (F_(4, 45)_ = 2.67; *p* = 0.0443), with significant interaction between the two factors (F_(12, 107)_ = 2.78; *p* = 0.0025). However, pairwise Dunnett’s multiple comparisons revealed no significant changes in AA levels across brain regions except for Hy (Figure 1C). These findings indicate that both eCBs were decreased prominently at 9 months, which is considered to be the middle age in mice, in specific brain regions. Thereafter, a moderate resurgence with increasing age was noticed in both the eCBs.

### 2.2. Prominent Reduction in Hippocampal DAGLα during the Early Middle-Age

We next investigated if the reduced hippocampal 2-AG levels were accompanied by changes in the levels of DAGLα. Quantifying the intensity DAGLα by antibody staining across hippocampal sub-regions separately revealed significant region-specific (F_(14, 770)_ = 114.597; *p* < 0.0001) and age-dependent (F_(4, 55)_ = 3.670; *p* = 0.0101) changes with no significant interaction (F_(56, 770)_ = 0.923; *p* = 0.6369). Pairwise Dunnett’s multiple comparisons revealed the most prominent decrease in DAGLα at 9 and 12 months in stratum pyramidale, stratum oriens, stratum radiatum, and stratum lacunosum-moleculare of CA1 and CA2 regions, whereas a less prominent, but significant, decrease in DAGLα protein at 9 and 12 months was only detected in the stratum radiatum of the CA3 region. Furthermore, a moderate decrease in DAGLα protein was also noticed at 18 months specifically in stratum oriens and stratum radiatum of the CA2 region (Figure 2A,B). Notably, the age-dependent changes in DAGLα protein were corresponding to the changes observed in 2-AG levels in the Hip.

### 2.3. Density of Hippocampal CB1 Increases during the Early Middle-Age

We next sought to determine the influence of age on CB1 density within the Hip. Quantitative analysis of CB1 immunoreactivity across the sub-regions of Hip showed significant region-specific (F_(14, 728)_ = 407.531; *p* < 0.0001) and age-dependent (F_(4, 52)_ = 13.249; *p* < 0.0001) changes with a strong interaction (F_(56, 728)_ = 1.556; *p* = 0.0070) between the two main factors. The pairwise Dunnett’s multiple comparisons showed a significant increase in CB1 protein at 9 months in the pyramidal layer of the CA1 region, whereas the other sub-regions of the Hip showed only a marginal increase. On the contrary, the CB1 protein decreased significantly at 12 months in all the sub-regions analyzed. In specific sub-regions, such as stratum lacunosum-moleculare of CA1/CA2, stratum lucidum of CA3, and the hilus, a marked reduction in CB1 protein was observed at both 12 and 18 months, whereas in other sub-regions, the decrease did not persist until 18 months (Figure 3A,B). These findings suggest that the notable increase in CB1 protein at 9 months could be a compensation to cope with the prominent decline in 2-AG levels.

### 2.4. DAGLα and CB1 Are Reduced with Age and in a Brain Region-Specific Manner in the Limbic Forebrain

To this end, we found a prominent reduction in 2-AG and its synthesizing enzyme DAGLα in the Hip during the middle age, as well as an increase in CB1 protein. We next analyzed DAGLα and CB1 proteins in other brain regions. Since the 2-AG levels were also reduced marginally during the middle age in CPu, we opted to analyze the age-related changes in DAGLα and CB1 proteins in this region along with the surrounding nucleus accumbens (Acb) and Cg. Interestingly, similar to Hip, DAGLα immunoreactivity across the three regions altered significantly in a brain region-specific (F_(2, 142)_ = 27.59; *p* < 0.0001) and an age-dependent manner (F_(4, 71)_ = 3.75; *p* = 0.008) with a significant interaction (F_(8, 142)_ = 6.65; *p* < 0.0001). However, unlike in the Hip, pairwise Dunnett’s multiple comparison tests showed that the earliest decrease in DAGLα in CPu was at 6 months, and thereafter, it further decreased gradually with advancing age in comparison with its levels at 2 months. Contrastingly, a prominent reduction in DAGLα protein was only seen at 18 months compared to the youngest group in both Acb and Cg (Figure 4A,C).

Analyzing the CB1 immunoreactivity across the same three regions also resulted in significant brain region-specific (F_(2, 132)_ = 315.68; *p* < 0.0001) and age-dependent (F_(4, 71)_ = 5.44; *p* = 0.0007) changes with a significant interaction (F_(8, 132)_ = 2.64; *p* = 0.0102) between the two factors. Pairwise Dunnett’s multiple comparisons revealed a significant reduction in CB1 protein in the CPu only at 12 months against 2 months. However, CB1 protein was significantly decreased in all age groups starting from 9 months in comparison to 2 months in Acb. In Cg, CB1 protein density was reduced significantly at 6, 9, and 12 months of age in contrast to the youngest group (Figure 4B,D).

### 2.5. Alterations in CB1-Specific Binding of [^3^H] CP55940 across Brain Regions

Despite the ability of immunofluorescence to visualize the distribution of a CB1 receptors anatomically by means of antibody detection of specific epitope of the receptor, absolute numerical quantification of receptor levels is not possible due to the lack of appropriate reference standards. As such, receptor levels can only be assessed based on the intensity of the staining with this technique. Moreover, it does not provide any information on the “functional sites” of a receptor capable of binding to its innate ligand, including receptor affinity. This information is, nonetheless, important, as changes in affinity could affect eCB signaling. As such, the CB1 autoradiographic binding technique was employed to complement the immunofluorescence assays as it enables the precise numerical determination of receptor sites capable of binding to its innate ligand. Although there were prominent age-related and brain region-specific changes in the density of CB1, as measured by immunofluorescence, it is not known how aging may influence the ligand-binding ability of the receptor across the brain regions. We, therefore, assessed the changes in the CB1-specific binding of [^3^H] CP55940, a potent cannabinoid agonist, with respect to age across brain regions (Figure 5).

The quantitative analysis of autoradiographic binding of [^3^H] CP55940 to CB1 showed significant brain region-specific alterations (F_(29, 982)_ = 130.94; *p* < 0.0001), with no significant age-dependent changes (F_(4, 35)_ = 0.927; *p* = 0.4594) or interaction (F_(116, 982)_ = 0.892; *p* = 0.7814). However, interestingly, pairwise Dunnett’s multiple comparisons revealed a prominent increase in CB1 binding specifically at 9 months against 2 months in the CA1–3 regions (Bregma −1.70). In addition, a nearly significant increase was observed in the dentate gyrus (*p* = 0.0588), again at 9 months compared to 2 months (Bregma −1.70). Similarly, CB1 binding also increased significantly in the whole Hip (Bregma −2.54) at 9 months in contrast to 2 months. On the other hand, the lateral entorhinal cortex (LEnt) showed a significant increase in CB1 binding at 9, 12 and 18 months in contrast to the youngest group. Conversely, the CB1 binding decreased significantly in the lateral globus pallidus, however, the decrease was only apparent at 6 months compared to 2 months and not in the later age groups. Similarly, the compact part and reticular parts of substantia nigra (SN) also showed significant decreases in CB1 binding, but only at 18 months against 2 months (Table 1). Together, these results indicate that the CB1 receptor binding ability is improved in the Hip and the adjacent LEnt to cope with the loss of 2-AG during the middle age.

## 3. Discussion

In the present study, we systematically characterized changes in the eCB system during the aging process in mice. Our findings demonstrate that both major eCBs, 2-AG and AEA, were affected with age in a brain region-specific manner. The 2-AG was most prominently reduced in the Hip and moderately in the CPu, whereas AEA was mostly decreased in the CPu as well as in the mPFC+Cg. Consistent with the changes in 2-AG, its synthesizing enzyme DAGLα was also down-regulated in the Hip. Conversely, the protein levels along with its ligand-binding density of its cognate receptor CB1 were prominently increased in the Hip. This increase may reflect a compensatory mechanism.

These data are consistent with previous results showing that the tone of the eCB system declines during aging. For example, our lab previously showed a prominent decline of 2-AG and its synthesizing enzyme DAGLα in the Hip at old age using two different mouse strains [9]. Although that study provided the first evidence of these changes at the time, the conclusions were drawn from a direct comparison of young (2 month) and old (12–15 month) mice. Furthermore, the alterations in the eCB tone during the aging process were not determined in that study. In the current study, we used five different age groups, ranging from 2 months to 18 months in order to characterize the influence of aging on the eCB system. We followed the age-categorization of Jackson Laboratory and other studies (https://www.jax.org/news-and-insights/jax-blog/2017/november/when-are-mice-considered-old# (accessed on 5 December 2021) [18,19]); therefore, 2-month-old mice were termed as adolescent, 6-month-old as adult, 9- and 12-month-old as middle aged and 18-month-old as old. The major finding of the current study is that the eCB tone is not linearly decreased with age, rather it is most affected during the middle age, and thereafter, it revived modestly.

Middle age is critical for several molecular and behavioral changes. Deep proteomic profiling of plasma proteome from young individuals to nonagenarians revealed undulating changes that indicate three waves of aging across the lifespan. They identified three distinct peaks during the fourth, seventh and eighth decades of life, indicating the number of proteins that are differentially expressed at that particular time window. Interestingly, among the most strongly altered proteins during the middle and old age, there are proteins that are associated with cognitive functions. These findings are an indication that the middle age is crucial for robust molecular changes [20]. Further, several rodent studies also reported prominent behavioral alterations during the middle age. For example, 2–3-, 4–5-, 6–7- and 8–12-month-old mice were tested for changes in motor function, locomotor activity, anxiety-like behavior, social behavior, depression-like behavior, and learning and memory functions with respect to age. Changes were observed in behaviors between individual age groups tested against the youngest group. Interestingly, for most behaviors, 8–12-month-old mice exhibited more pronounced changes than the other age groups compared to the youngest group [21]. A more recent and similar study also reported age-related changes in spatial learning and memory in C57BL/6J mice [22]. However, this study tested the learning and memory functions at 2, 4, 6, 9 and 12 months of age and reported a gradual decline in these performances only from 6 months of age [22]. In another study, deficits were observed in the Hip-dependent contextual fear conditioning task in middle-aged mice. Afterhyperpolarization is a mechanism generally engaged during the learning process. Although basal neuronal excitability seems to be comparable in both young and middle-aged mice, learning-related modulation of the post-burst afterhyperpolarization was impaired in CA1 neurons in middle-aged mice [23]. Emotional behaviors were also tested in rats at 3 (young) and 12 months (middle-age) of age. It was shown that middle-aged rats were more anxious, depressed, and displayed aversive memory impairments [24].

Importantly, age affects brain on an area-specific manner: reduction in white-matter and cell loss is reported in the cortex, hippocampus, putamen, thalamus or striatum, whereas other brain regions remain largely preserved during physiological aging [25]. Thus, the cerebellum, which is a major brain area with an exceptionally high CB1 receptor expression [26], was not investigated in the present study.

CB1 mediates two types of signaling cascades in the hippocampus based on its location. CB1 on the Schaffer commissural projections mediates phosphorylation of Munc18-1 through ERK1/2 signaling to depress the transmitter release [27]. Most CB1 agonists are capable of activating this signaling cascade. However, 2-AG mediated CB1 signaling on the lateral perforant path terminals entails β1-integrin to effect synaptic potentiation by stably enhancing the transmitter release. This signaling cascade is explicitly engaged during the lateral perforant path-dependent learning [28,29]. Interestingly, perforant path projections of layer II entorhinal cortex neurons deteriorate during middle age, accompanied by prominently impaired long-term potentiation (LTP) in the medial and lateral perforant paths in both mice and rats. Accordingly, middle-aged mice also performed poorly in the lateral perforant path-dependent episodic memory task. Importantly, disrupted eCB signaling due to changes in 2-AG levels were attributed to the deterioration of later perforant path LTP during the middle age. Although data for basal 2-AG levels among the age groups were lacking, it was shown that increasing 2-AG levels by inducing DAGLα expression using physostigmine or by blocking the breakdown of 2-AG using JZL184 restored the lateral perforant path LTP [30]. In line with the above findings, we also show here that 2-AG levels were prominently affected during the early middle-age in both Hip and CPu. The findings from *Daglα* knock out (KO) mice have clearly shown that a loss of 2-AG adversely affects the emotional state in mice, resulting in enhanced anxiety, stress, fear responses, sociability deficits, and impaired exploratory drive [3,31]. Moreover, the fact that middle-aged C57BL/6J mice also exhibit the above stated behavioral deficits while 2-AG signaling is most affected is quite fitting together.

Here, we show that CB1 protein levels were first slightly increasing in the 2–9 month age interval in most sub-regions of the Hip and dropped significantly at 12 months in all the sub-regions of the Hip. Further, we also observed a CB1 protein decrease in CPu (at 12 months), Acb (at 9, 12 and 18 months), and Cg (at 6, 9 and 12 months). These undulating changes in CB1 protein levels, specifically during the middle age in the sub-regions of the Hip, may constitute a compensation for the 2-AG loss. Although, the molecular mechanisms underlying this phenomenon are not clear, the compensation seems to be specific to the Hip and not for other regions tested. In agreement with this, early studies also reported a decrease in *Cb1* mRNA levels in aged rats (>24 months) against young (3 months) rats. It was shown that *Cb1* mRNA levels were specifically decreased in CPu, and to a small extent, in the sub-regions of Hip (CA1, CA2, CA3, and DG) [32,33]. Age-related changes in CB1 protein levels were also compared between 4- and 24-month-old rats using Western blotting. Interestingly, the study reported an increase in CB1 protein levels in entorhinal and temporal cortices, and a decrease in the postrhinal cortex, whereas there were no changes in the Hip or its sub-regions [34]. The above studies unfortunately lack data from other age groups. In human muscle, the CB1 protein levels seem to increase with age when compared between young (20–27 years) and old (≥65 years) individuals, and CB1 expression seems to increase with resistance exercise in older individuals [35]. Mice subjected to voluntary exercise for eight days exhibited increased eCB signaling in the Hip, accompanied by an increased progenitor cell proliferation within the DG. Eight days of free access to running wheels also increased the agonist binding site density of CB1 receptors within the Hip [36]. Aerobic exercise with moderate intensity induced antinociception in rats that was mediated through CB1. Exercise also induced the expression and activation of CB1 in rat brains (periaqueductal gray matter) assessed using Western blotting and immunofluorescence. In addition, exercise also elevated the plasma levels of both AEA and 2-AG [37]. This is an indication that a healthy lifestyle by means of regular exercise could delay or postpone the middle-age crisis in the eCB system.

Aging has been reported to alter binding of CB1 in the brain. In aged rats (>24 months), [^3^H]-WIN 55,212-2 specific binding of CB1 was shown to be decreased in most basal ganglia regions (entopeduncular nucleus, SN and CPu) other than globus pallidus [32]. The same group also analyzed CB1 receptor binding of [^3^H]-WIN 55,212-2 in the sub-regions of Hip and showed similar binding levels in both young and aged rats, except for the DG, where the binding was increased in aged rats. Further, CB1 binding was significantly decreased in the Hy, whereas similar binding levels were detected in the limbic structures (nucleus accumbens, septum nuclei, and basolateral amygdaloid nucleus) of young and aged rats [33]. In line with the above findings, [^3^H] CP55940 specific binding of CB1 was significantly decreased in the SN at the age of 18 months. Furthermore, CB1 binding in the dorsal Hip (CA1, CA2, CA3 and DG) and the ventral Hip was increased specifically at 9 months, but in LEnt, CB1 binding increased at 9, 12, and 18 months. However, unlike CB1 protein levels in the sub-regions of the dorsal Hip at 12 months, no prominent decrease in CB1 binding was detected at 12 months and later. The discrepancy between CB1 protein levels after 9 months of age as measured by immunofluorescence, and CB1 binding as measured by receptor autoradiography, may be due to the different experimental techniques. Immunofluorescence measures epitope of a receptor, whereas receptor binding measures the interaction between the receptor binding site and the ligand. Changes in binding levels reflect not only altered receptor numbers, but also changes in affinity (Kd). Indeed, CB1 is known to undergo positive allosteric modulation by endogenous anti-inflammatory lipids, such as lipoxin A4 [38,39]. As such, changes in CB1 binding detected in 9-month-old mice may reflect aging-dependent changes in the levels of endogenous allosteric modulators of CB1 which inadvertently would impact CB1 affinity and, hence, binding levels. Nonetheless, age-related changes in the receptor number (Bmax), but not in Kd were also reported for other receptor systems in the brain, such as dopamine D2/D3 receptors [40]. Hence, the changes in binding in our study most likely represent changes in receptor number rather than changes in affinity.

The density and signaling properties of CB1 vary based on the cell type in the Hip of adult mice. GABAergic neurons express higher amounts of CB1 receptors compared to glutamatergic neurons, but these CB1 receptors are less efficiently coupled to G-protein signaling than their glutamatergic counter parts [41]. Although we have not determined this in the current study, it is necessary to understand how cell-type specific CB1, its ligand-binding affinities, and downstream signaling are affected with age. As such, future work is warranted, focusing on the consequences of aging on CB1 function and signaling with the use of a combination of several techniques, including CB1 agonist [^35^S]GTPγS binding, intracellular ^++^Ca imaging, signaling, and electrophysiology.

In conclusion, our observations indicate that the eCB system is most affected during the middle age in a brain region-specific manner. The 2-AG levels were mostly affected in the Hip and moderately in CPu. While AEA levels were not altered in Hip, they were mainly affected in CPu as well as in mPFC+Cg. Although CB1 protein levels prominently decreased at 12 and 18 months of age in most sub-regions of the Hip, CB1 binding was unaltered at this age in these regions. Taken together, the middle-age crisis in the eCB signaling corresponds well with the onset of neuroinflammatory glial activity and cognitive deficits in mice. We now hypothesize that late middle-age is the time period when a therapy based on the activation of the cannabinoid system has the highest efficacy to prevent cognitive aging and pathologies related to brain aging.

## 4. Materials and Methods

### 4.1. Animals

A total of three independent cohorts of C57BL/6J male mice (age groups: 2, 6, 9, 12 and 18 months old) were purchased separately in due course from Janvier, France. For each cohort, mice were group housed (five per cage) and acclimatized for at least two weeks, under normal light/dark cycle at the House of Experimental Therapy, Medical Faculty, University of Bonn, before the brain samples were collected. All experiments were approved by the North Rhine-Westphalia State Environment Agency (LANUV, Landesamt fuer Natur, Umwelt und Verbraucherschutz, license nr: AZ: 81–02.04.2019.A322) and were performed in accordance with the relevant guidelines and regulations.

### 4.2. Endocannabinoids Extraction

Cohort 1, *n* = 8–10 mice per age group, were sacrificed via cervical dislocation and the brains were isolated rapidly and placed in ice-cold PBS. Hy, Hip, CPu and mPFC+Cg were simultaneously dissected and flash-frozen in liquid nitrogen. The whole procedure was performed within five minutes of killing the mice to minimize post-mortem changes in the eCB levels. The extraction procedure and the quantitative profiling of eCBs using the liquid chromatography (LC)/multiple reaction monitoring (MRM) was performed as described previously [9,42]. In brief, frozen brain samples were weighed and transferred on dry ice to Precellys^®^ tubes containing steel balls (2.8 mm). Ethylacetate/*N*-hexane (9:1) and 0.1 M formic acid spiked with deuterated internal standards (AEA-d_4_; 2-AG-d_5_; AA-d_8_) (BIOMOL Research Laboratories Inc. (Plymouth Meeting, PA, USA)) were then added to the tubes. For eCB extraction, samples were then homogenized in Precellys^®^ Evolution (1 cycle for 20 s/6000 rpm) (Peqlab, Erlangen, Germany) and eventually centrifuged to separate the upper organic phase (5000× *g*/4 °C for 10 min). The organic phase was carefully collected into deep-well plates and evaporated under a gentle stream of nitrogen at 37 °C and reconstituted with 50 μL of water/acetonitrile (1:1) for eCB measurement.

### 4.3. LC/MS Quantitative Analyses

A total of 20 μL of the solution of extracted eCBs were injected and separated on a Phenomenex Luna 2.5 μm C18(2)-HST column, 100 × 2 mm, combined with a pre-column (C18, 4 × 2 mm; Phenomenex, Aschaffenburg, Germany), by increasing acetonitrile containing 0.1% formic acid over 2 min from 55 to 90%, and maintaining it at 90% for 5.5 min. The separated eCBs were flow-through analyzed using MRM on a 5500 QTrap triple-quadrupole linear ion trap mass spectrometer equipped with a Turbo V Ion Source (AB SCIEX, Darmstadt, Germany). Positive and negative ions were simultaneously analyzed using the ‘positive-negative-switching’ mode. The following MRM transitions were monitored for quantification of eCBs: AEA, *m*/*z* 348.3 to *m*/*z* 62.3; 2-AG, *m*/*z* 379.1 to *m*/*z* 287.2; AA, *m*/*z* 303.05 to *m*/*z* 259.1. Calibration solutions were prepared using commercially available standards of high purity, spiked with a mixture of deuterated eCBs and run in triplicates. Quantification of eCBs was carried out using Analyst 1.6.1 software (AB SCIEX, Darmstadt, Germany). The eCB concentrations were normalized to protein content (for tissues) measured by BCA and to serum volume.

### 4.4. Immunofluorescence

Cohort 2, *n* = 4 mice per age group, were decapitated and whole brains were instantly frozen for 30 s in isopentane cooled below −20 °C and immediately transferred to −80 °C for later use. Coronal sections (20 μm thick) containing Cg, CPu, Acb, and Hip were sliced using cryostat (Leica CM 3050, Leica Microsystems, Heidelberg, Germany), mounted on the Superfrost^®^ Plus glass slides (Karl Roth, Karlsruhe, Germany), and stored at −80 °C for further use. At the time of staining, frozen slides were warmed to room temperature and briefly washed in PBS for 5 min and the sections were post-fixed in 4% paraformaldehyde prepared in fresh PBS for 30 min. After washing twice in PBS for 5 min, sections were permeabilized with 0.3% Triton X-100 for 10 min. Post incubation, sections were washed with PBS for 5 min and incubated in blocking solution (5% normal goat serum and 0.2% Triton X-100 prepared in PBS) for 1 h at RT to block the non-specific binding. Following a brief 5 min wash, sections were incubated for 24 h at 4 °C with either rabbit anti-CB1, rabbit anti-DAGLα, and guinea pig anti-NeuN primary antibodies at a dilution of 1:500 (CB1-Rb-Af380, DAGLα-Rb-Af380; Frontier Institute Co. Ltd., Hokkaido, Japan; NeuN-266004; Synaptic Systems GmbH, Gottingen, Germany) prepared in 0.5% BSA and 0.05% Triton X-100. Post incubation, sections were washed three times for 5 min each with PBS, and then labeled with secondary antibodies (Donkey anti-Rabbit Alexa Fluor™ 647, #A-31573; Goat anti-Guniea Pig Alexa Fluor™ 568, #A-11075, Life Technologies, Darmstadt, Germany) at 1:2000 dilution. Sections were then washed three times for 5 min each with PBS, and after a brief wash with MilliQ water, slides were cover slipped with DAPI mounting medium (DAPI Fluoromount-G^®^, Southern Biotechnology, Birmingham, AL, USA). Sections were allowed to dry at RT and then stored at 4 °C until imaging. The specificity of both anti-CB1 and anti-DAGLα antibodies were assessed using the sections from CB1 KO and DAGLα KO mice (Appendix A). Images were obtained using LSM SP8 confocal microscope (Leica, Wetzler, Germany) with 40× water immersion objective. The whole Hip was imaged for the quantification of DAGLα or CB1 proteins, and the sub-regions were traced and quantified using Fiji software. DAGLα or CB1 intensities were quantified from at least 3–4 areas from each sub-region and were represented as individual data points. For CPu, Acb, and Cg, 3–4 images were captured from each region to quantify DAGLα or CB1 intensities and were represented as individual data points.

### 4.5. Autoradiographic Binding of CB1 Receptor

Cohort 3, *n* = 8 mice per age group, were sacrificed through cervical dislocation and whole brains were instantly frozen for 30 s in isopentane cooled below −20 °C and immediately transferred to −80 °C for later use. Serial coronal sections (20 μm thick) from Bregma 2.46 to −2.54 were sliced and mounted sequentially on two sets of pre-labeled, 1% gelatin/chrome alum coated, Superfrost^®^ Plus glass slides. The slides were carefully placed in a slide holder box which was then placed in an air-tight container filled with Drierite™ with indicator (#238988, Sigma-Aldrich, St. Luis, MI, USA) at the base to keep the slides dry. The air-tight box was then moved into 4 °C for at least 2 h and then eventually moved into −20 °C for storage until use. Two sets of slides having adjacent sections, one for the CB1-specific total binding and the other for non-specific binding, were used for the autoradiography experiment.

CB1 binding was carried out in accordance to protocol developed by [43]. Microscope slides containing sections (total and NSB) were rinsed in a pre-incubation buffer of 5 mM Tris (Sigma, Poole, UK), pH 7.4 at room temperature for 10 min. Slides used to determine total binding were incubated in 5 mM Tris, 5% bovine serum albumin buffer containing 10 nM [^3^H] CP-55,940 (Perkin Elmer, Waltham, MA; USA) at RT, pH 7.4 for 2.5 h. Adjacent slides used for non-specific binding were incubated in 5 mM Tris, 5% bovine serum albumin (Sigma, Poole, UK) containing 10 nM [^3^H] CP-55,940 and 10μM CP-55,940 (Sigma, Poole, UK) at RT, pH 7.4 for 2.5 h. Following the incubation, all sections were rinsed twice with 50 mM Tris buffer containing 1% bovine serum albumin, pH 7.4 for 2 h each time at 0 °C. The sections were then dipped in ice-cold distilled water and dried in cold air for 2 h. They were then put to dry in airtight containers containing a layer of anhydrous calcium sulphate (Drierite-BDH chemicals, Dorset, UK) for a minimum of 7 days. All brain sections were processed together to avoid any inter experimental variation.

The slides (total and NSB) from all experimental groups were placed side-by-side to Kodak MR-1 films in hyper cassettes together with autoradiographic [^3^H] microscales of known radioactive concentration cross-calibrated to protein density. Sections from all age groups were arranged in parallel, processed, and apposed to the same film simultaneously to avoid inter-experimental variations. Following 6 weeks of apposition, film development took place in a dark room with red filter light. The films were immersed for 2 min in a tray containing 50% Kodak D19 developed; next, for 30 s, in a tray containing distilled water and 3 drops of glacial acetic solution and lastly for 2 min in a tray containing Kodak rapid fix solution.

Quantitative analysis of autoradiographic films was carried out by video-based, computerized densitometry using an MCID image analyzer, as previously described by Kitchen and co-workers [44]. Optical density values were quantified from the [^3^H]-microscale standards of known radioactive concentration and were entered into a calibration table on MCID. Specific binding was calculated by subtracting non-specific binding from total binding and expressed as fmol/mg tissue equivalents. Brain structures were identified by reference to the Franklin and Paxinos (1997) mouse atlas.

### 4.6. Data Analysis

Data were analyzed using R software package (nlme) or GraphPad Prism 9 (STAT*CON* GmbH, Witzenhausen, Germany), and are represented as either mean ± SEM (Figure 1 and Table 1) or boxplots, with whiskers representing minimum and maximum values (Figure 2, Figure 3 and Figure 4). The values for eCB levels, Daglα and CB1 signal intensities in the Hip, CPu, Acb and Cg were log-transformed (natural logarithm). The untransformed data were not normally distributed and highly right-skewed, whereas the log-transformed data approached a normal distribution and improved the model fit. There is an ongoing debate in the scientific community about the optimal strategy when data are not normally distributed—data transformation or non-parametric analysis. However, non-parametric tests cannot be used if two or more independent variables are considered (in our case “brain region” and “age”). In our study, we aim to assess the effect of “aging” as well as the “aging x brain region” interaction in order to investigate whether any changes in levels of the endocannabinoid system (CB1 receptor or endogenous ligand) induced by aging are brain specific or not (i.e., if changes are dependent on brain region). As such, to achieve our aim, we used a linear mixed-effects model. As the data were not normally distributed, we first log-transformed our data and subsequently implemented the mixed model analysis followed by pairwise comparisons Dunnett’s test. In all the experiments, the significant changes depicted among the age groups were always compared against the youngest age group (2-month-old) as this was our control group. Significant outliers were removed using the ROUT method and the alpha level of statistical significance was set to 0.05. The group size for the individual dataset is given in the respective figure legend.

## Figures and Tables

**Figure 1 ijms-23-10254-f001:**
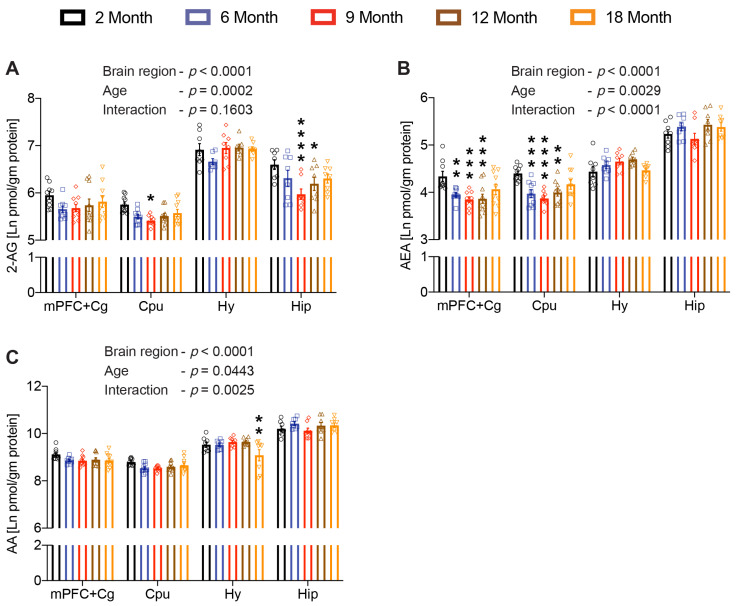
Age-related changes in the endocannabinoids. Brain 2-AG (**A**), AEA (**B**) and AA (**C**) levels were analyzed in medial prefrontal cortex including the cingulate cortex (mPFC+Cg), striatum (CPu), hypothalamus (Hy), and hippocampus (Hip) of 2-, 6-, 9-, 12- and 18-month-old mice. Both 2-AG and AEA levels were significantly altered in a brain region-specific manner across the age groups, whereas AA levels were only altered specifically in the hypothalamus of aged mice (*n* = 7–10 mice per age group). Data are represented as mean ± SEM, * *p* < 0.05; ** *p* < 0.01; *** *p* < 0.001; **** *p* < 0.0001; linear mixed model analysis followed by Dunnett’s multiple comparison test. *p*-values refer to pairwise tests against the 2-month-old group.

**Figure 2 ijms-23-10254-f002:**
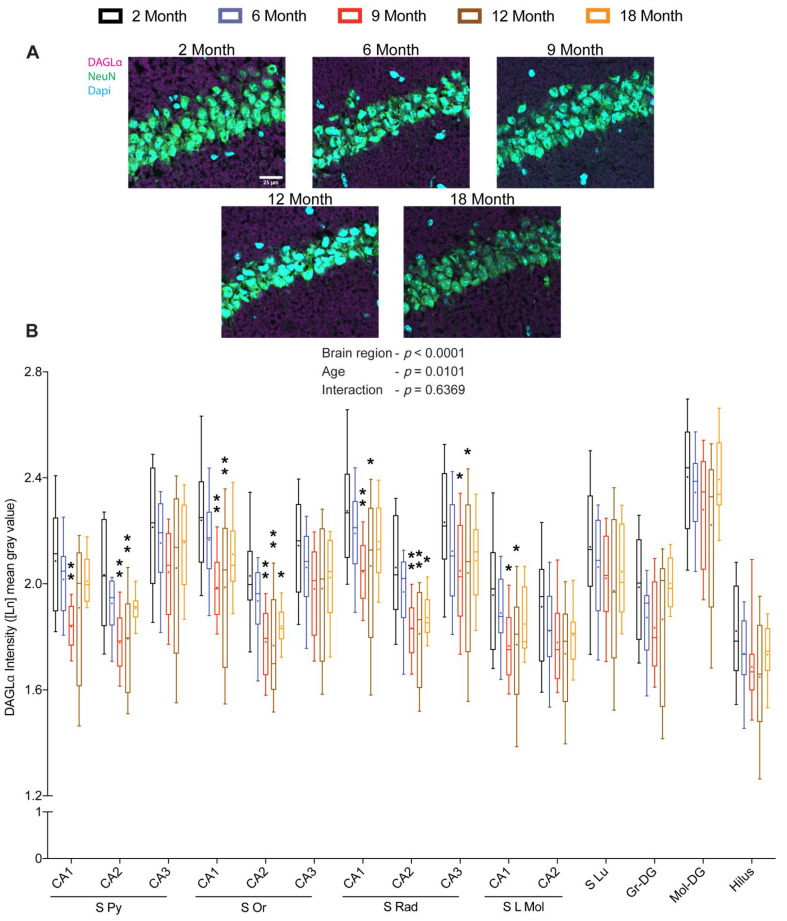
Effect of age on DAGLα protein levels within hippocampus. Coronal brain sections of 2-, 6-, 9-, 12- and 18-month-old mice immunostained for DAGLα. Representative images depicting for DAGLα immunoreactivity in the CA1 region (scale bar = 25 μm) (**A**). Quantification of DAGLα immunoreactivity across hippocampal sub-regions revealed significant changes in DAGLα protein levels with respect to age and brain region (**B**). (*n* = 12 images were analyzed from four mice per age group). Whiskers represent minimum and maximum values; + symbol within the box indicates mean; * *p* < 0.05; ** *p* < 0.01; linear mixed model analysis followed by Dunnett´s multiple comparison test. *p*-values refer to pairwise tests against the 2-month-old group.

**Figure 3 ijms-23-10254-f003:**
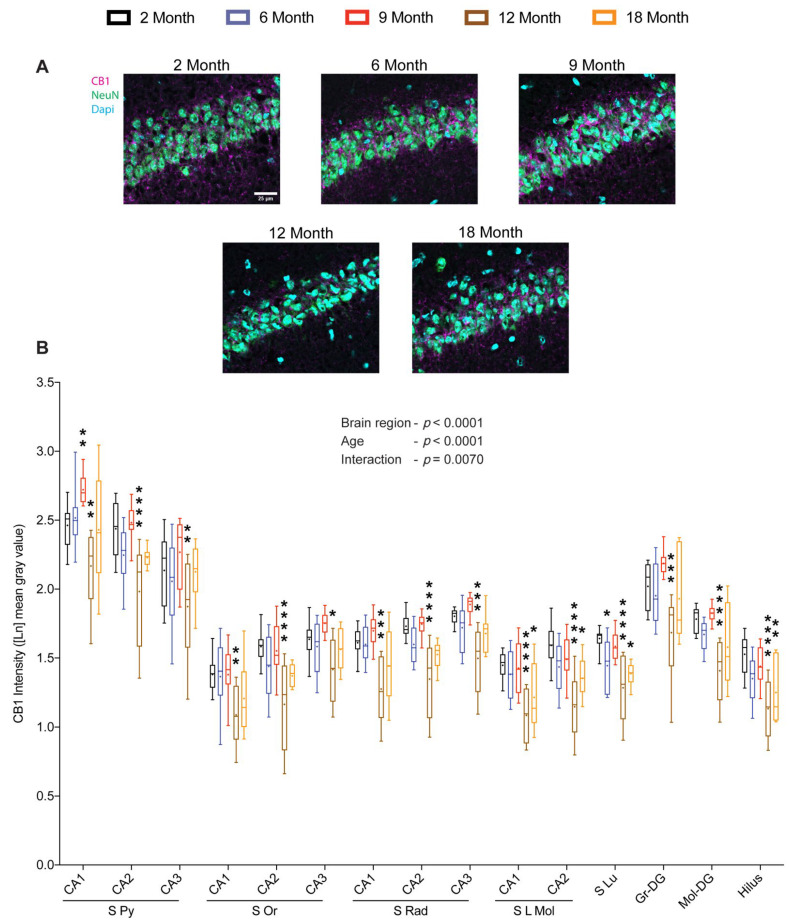
Effect of age on CB1 protein levels in the hippocampus. Coronal brain sections of 2-, 6-, 9-, 12- and 18-month-old mice were immunostained for CB1 receptor. Representative images depicting CB1 immunoreactivity in the CA1 region (scale bar = 25 μm) (**A**). Quantification of CB1 immunoreactivity across the sub-regions of the hippocampus revealed significant changes in CB1 protein levels with respect to age and brain region (**B**). (*n* = 9–12 images were analyzed from 3–4 mice per age group). Whiskers represent minimum and maximum values; + symbol within the box indicates mean; * *p* < 0.05; ** *p* < 0.01; *** *p* < 0.001; **** *p* < 0.0001; linear mixed model analysis followed by Dunnett´s multiple comparison test. *p*-values refer to pairwise tests against the 2-month-old group.

**Figure 4 ijms-23-10254-f004:**
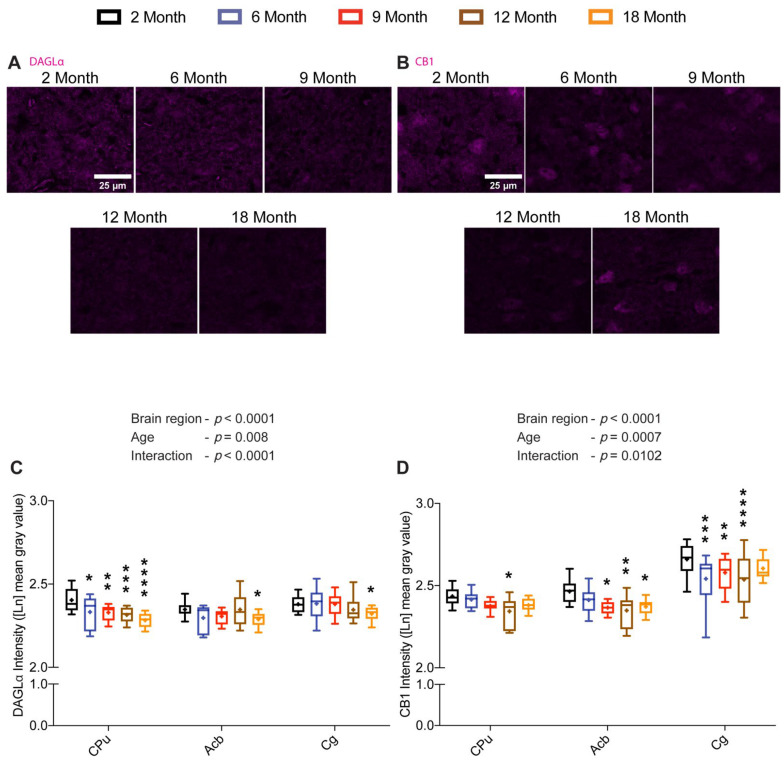
Age-related changes in DAGLα and CB1 protein levels in the limbic forebrain. Brain sections of 2-, 6-, 9-, 12- and 18-month-old mice were immunostained for DAGLα and CB1. Representative images depicting DAGLα and CB1 immunoreactivity in the striatum (scale bar = 25 μm) (**A**,**B**). Quantification of DAGLα (**C**) and CB1 (**D**) immunoreactivity in different limbic forebrain regions showed significant changes in DAGLα and CB1 protein levels in a region-specific manner with respect to age (*n* = 12–16 images were analyzed per region from 3–4 mice per age group). Whiskers represent minimum and maximum values; + symbol within the box indicates mean; * *p* < 0.05; ** *p* < 0.01; *** *p* < 0.001; **** *p* < 0.0001; linear mixed model analysis followed by Dunnett´s multiple comparison test. *p*-values refer to pairwise tests against the 2-month-old group.

**Figure 5 ijms-23-10254-f005:**
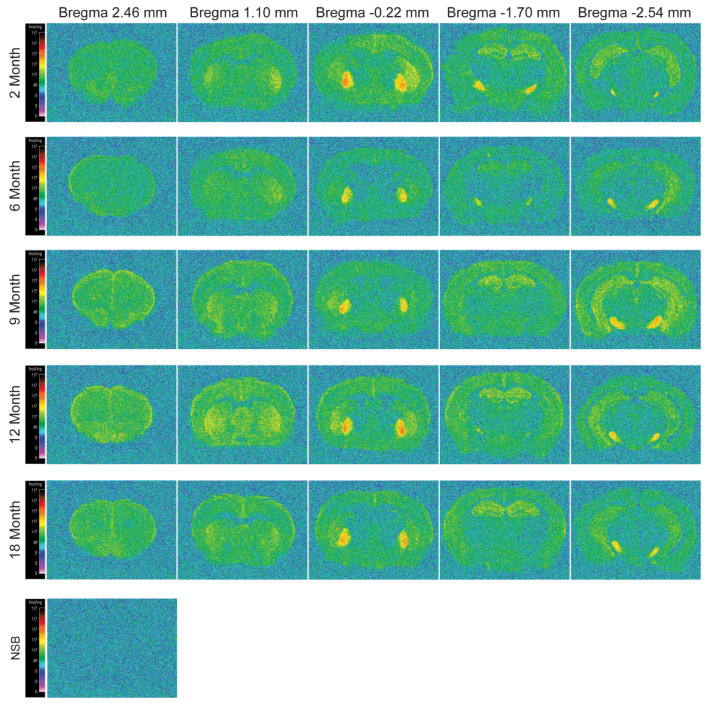
Computer-enhanced color autoradiograms of [^3^H] CP55940 (10nM) binding to CB1 receptors in coronal brain sections of 2-, 6-, 9-, 12- and 18-month-old C57BL/6J mice. Coronal brain sections are represented at the level of the prelimbic cortex (column 1; Bregma +2.46 mm), striatum (column 2; Bregma +1.10 mm), lateral geniculate nucleus (column 3; Bregma −0.22 mm), hippocampus (column 4; Bregma −1.70 mm) and lateral entorhinal cortex (column 5; Bregma −2.54 mm). Adjacent sections were incubated with 10 μm CP55940 to determine non-specific binding (NSB). Binding levels are represented using pseudo-color interpretation of black and white film images in fmol/mg of tissue equivalent.

**Table 1 ijms-23-10254-t001:** Influence of age on CB1 specific binding in various brain regions.

		Age Groups				
Region	Bregma (mm)	2 Month	6 Month	9 Month	12 Month	18 Month
M1 + M2	+2.46	70.53 ± 4.47	65.68 ± 7.71	78.99 ± 8.09	74.03 ± 8.18	78.62 ± 8.57
LO + VO	+2.46	65.16 ± 4.67	63.45 ± 6.71	71.82 ± 5.59	69.89 ± 6.49	71.89 ± 7.16
AON	+2.46	82.67 ± 5.59	75.04 ± 9.33	89.84 ± 8.49	88.57 ± 6.93	88.50 ± 7.86
PrL	+2.46	75.47 ± 4.8	68.89 ± 8.81	82.83 ± 8.26	78.84 ± 8.65	82.89 ± 8.51
M1 + M2	+1.10	72.79 ± 4.86	67.65 ± 5.77	83.97 ± 6.19	80.83 ± 6.93	82.04 ± 7.69
S1	+1.10	62.84 ± 4.84	58.31 ± 5.16	68.54 ± 4.51	64.85 ± 6.32	69.15 ± 6.78
Cg1 + Cg2	+1.10	76.20 ± 4.93	69.36 ± 7.00	85.32 ± 6.27	84.04 ± 7.76	87.87 ± 9.12
Cpu	+1.10	92.17 ± 4.35	79.79 ± 6.52	91.20 ± 6.49	84.91 ± 9.06	84.97 ± 8.2
AcbSh	+1.10	81.79 ± 9.01	65.55 ± 5.33	79.83 ± 6.83	83.74 ± 7.57	74.87 ± 7.40
AcbC	+1.10	88.73 ± 10.18	73.92 ± 7.56	88.74 ± 10.29	87.69 ± 7.49	83.00 ± 9.32
DEn	+1.10	69.02 ± 4.61	60.56 ± 3.75	73.67 ± 5.18	81.51 ± 6.95	78.72 ± 7.0
LS	+1.10	74.66 ± 5.15	67.10 ± 6.47	83.24 ± 8.43	82.31 ± 8.78	81.99 ± 7.84
MS	+1.10	73.20 ± 7.61	62.68 ± 4.72	73.88 ± 6.35	77.96 ± 9.95	70.53 ± 7.76
Tu	+1.10	71.23 ± 5.42	58.49 ± 3.97	72.06 ± 5.9	71.30 ± 7.45	67.05 ± 7.18
PGP	−0.22	150.12 ± 10.53	127.42 ± 10.82 **	144.26 ± 7.22	145.05 ± 11.2	135.72 ± 8.98
CA1	−1.70	117.27 ± 6.18	117.33 ± 7.28	138.41 ± 8.39 *	126.16 ± 11.49	126.99 ± 10.38
CA2	−1.70	119.58 ± 5.8	123.30 ± 9.9	140.8 ± 6.9 *	129.09 ± 11.56	135.00 ± 10.42
CA3	−1.70	129.31 ± 7.40	138.34 ± 12.30	155.03 ± 8.62 **	138.37 ± 12.21	138.82 ± 11.18
DG	−1.70	129.00 ± 8.24	130.27 ± 10.93	146.28 ± 8.3 a	134.75 ± 13.81	135.34 ± 11.4
Hy	−1.70	58.29 ± 4.92	51.24 ± 3.84	54.89 ± 3.37	55.41 ± 6.92	53.83 ± 6.04
Amy	−1.70	56.29 ± 3.1	53.74 ± 4.54	62.95 ± 4.30	65.14 ± 7.03	65.24 ± 6.75
BLA	−1.70	60.31 ± 4.65	56.21 ± 4.56	65.32 ± 3.77	71.64 ± 8.21	66.34 ± 6.58
BMA	−1.70	53.77 ± 3.37	50.74 ± 5.21	58.33 ± 5.55	58.65 ± 6.08	62.27 ± 7.21
Pir	−1.70	53.43 ± 3.88	50.01 ± 4.07	58.84 ± 4.51	62.80 ± 5.8	63.92 ± 7.17
S1 + S2	−1.70	57.68 ± 4.28	56.02 ± 4.10	66.59 ± 3.43	67.02 ± 7.60	65.76 ± 7.12
Th	−1.70	38.63 ± 4.30	34.26 ± 3.11	40.13 ± 3.56	37.04 ± 4.47	38.34 ± 5.49
SNC + SNR	−2.54	130.56 ± 10.19	135.85 ± 8.34	132.91 ± 13.69	98.15 ± 17.05	105.87 ± 14.11 **
PAG	−2.54	62.23 ± 4.82	61.40 ± 3.94	68.80 ± 6.15	54.64 ± 4.93	59.21 ± 7.65
Hip	−2.54	80.25 ± 4.48	87.39 ± 4.99	102.23 ± 5.49 **	82.47 ± 8.14	93.38 ± 6.22
LEnt	−2.54	55.31 ± 2.38	60.17 ± 5.14	72.36 ± 4.69 *	74.75 ± 5.34 *	73.27 ± 5.84 *

Quantitative CB1 receptor autoradiographic binding with [^3^H] CP55940 in brain sections of C57BL/6J strain of mice at 2, 6, 9, 12 and 18 months of age (*n* = 7–8 mice per age group). Specific binding of [^3^H] CP55940 (fmol/mg tissue) from left and right brain hemispheres. Data are expressed as mean ± SEM, a = 0.0588, * *p* < 0.05, ** *p* < 0.01; linear mixed-effects model analysis followed by Dunnett´s multiple comparison test. *p*-values refer to pairwise tests against the 2-month-old group. M1 + M2, Motor cortex (primary, secondary); LO + VO, Orbital cortex (lateral and ventral); AON, anterior olfactory nucleus (lateral, medial and ventral); PrL, prelimbic cortex; S1 somatosensory cortex (primary); Cg1 + Cg2, cingulate cortex (area 1 and area 2); CPu, caudate putamen; AcbSh, nucleus accumbens shell; AcbC, nucleus accumbens core; DEn, dorsal endopiriform nucleus; LS, lateral septal nucleus (dorsal, ventral and stripe of the striatum); MS, medial septal nucleus; Tu, olfactory tubercle; LGP, lateral globus pallidus; CA1, cornu ammonis 1; CA2, cornu ammonis 2; CA3, cornu ammonis 3; DG, dentate gyrus; Hy, hypothalamus; Amy, amygdala; BLA, basolateral amygdaloid nucleus, anterior part; BMA, basomedial amygdaloid nucleus, anterior part; Pir, piriform cortex; S1 + S2, somatosensory cortex (primary and secondary); Th, thalamus; SNC + SNR, substantia nigra (pars compacta and reticularis); PAG, periaqueductal gray; Hip, hippocampus; Lent, lateral entorhinal cortex.

## Data Availability

Data presented in this article are available on a reasonable request from the corresponding author.

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
