# Peer review of "Dynamic Changes in the Endocannabinoid System during the Aging Process: Focus on the Middle-Age Crisis"

_ijms, 2022, doi:10.3390/ijms231810254_

Round 1

Reviewer 1 Report

Review of “Dynamic changes in the endocannabinoid system during the aging process: focus on the middle age crisis”

The authors examined the alteration of the endocannabinoid system in mice as a function of aging.  The endocannabinoids 2-arachidonyl glycerol (2-AG) and anandamide (AEA) were extracted from specific dissected regions of the brain for cohorts (n=8-10) of mice per age group in the 2, 6, 9, 12, and 18 month age ranges. Quantification of endocannabinoids were acquired using triple-quad mass spectrometry in the selective reaction monitoring mode. Cannabinoid receptor CB1 expression and the 2-AG synthesizing enzyme DAGL-a were monitored using immunofluorescence con additional cohorts (n=4) in each age group. In addition, the binding efficiency of the specific CB1 agonist [H3]-CP-55,940 was also monitored for cohorts (n=8) for each age group using autoradiography. Overall, the results indicate that the endocannabinoid system is affected the most during the middle age (9-12 months) and in a brain region-specific manner.

This manuscript is well written, and the results/discussion presented in manner that is readily followed. The material and methods section were clearly written and also reasonably easy to follow. Figures are well presented, and the legends clearly describe the content. The introduction and discussion were well referenced and support the conclusions presented here.  The comparison to the behavioral alterations noted in reference 20 and 21 are particularly compelling.

The reviewer highly recommends the publication of this manuscript in its current form.

Author Response

We thank the reviewer for her/his positive comments. These are greatly appreciated and we fully agree with it. There were no concerns to be addressed

Reviewer 2 Report

This study investigated the changes in endocannabinoid levels in parallel to the immunostaining analysis of the 2-AG synthesizing enzyme DAGL and the CB1 receptor (also analyzed with binding autoradiography) at 5 different ages in mice: from adolescent animals (2 month-old) to aged individuals (18 month-old). This is an interesting idea supported by previous data from the same group indicating that changes in specific elements of the endocannabinoid signaling may have a role in certain age-related behavioral impairments, something found predominantly in the hippocampus. Despite this interest, there are some questions that may require changes in the text and even the development of additional measures. They are listed below:

1.       An important concern is the transformation of data to log scales. Authors have explained that this is necessary to approach the data to normal distributions, but, in my opinion, data should have been better assessed with a non-parametric method (Kruskal-Wallis) instead such artificial transformation that may generate some false differences.

2.       The use of ligand-binding autoradiography is correct, but it does not provide any information on receptor functionality, as authors indicated to support this use. For such purpose, it would have been more adequate to use agonist-stimulated GTPgS binding autoradiography or procedures to measure receptor intracellular signaling. 

3.       Extending the above question, autoradiography (ligand binding or agonist-stimulated GTPgS binding) has not the necessary anatomical resolution to detect specific signal in so small brain areas as some of the small nuclei included in Table 1. These measures may be, in some cases, a mere artifact. Authors should be cautious with these data.  

4.       One important CNS structure of high interest in this type of studies is the cerebellum. Authors should also add data of endocannabinoid levels, DAGL and CB1 immunostaining and CB1 binding autoradiography in this structure, in fact, it is possible that this area may be already visible for analysis in some of the procedures used. 

5.       Lines 44-48: Authors should differentiate the enzymes involved in the generation of endocannabinoids to those involved in their degradation. 

6.       Line 106: In terms of the metabolism of endocannabinoids, it is not correct to say that arachidonic acid is the precursor of 2-AG and AEA, as both are synthesized from more complex lipids, of course containing arachidonic acid. 

7.       Line 258: AEA, not EAE.

Author Response

Comment 1: The reviewer expressed concern over the transformation of data to log scales. Although the reviewer says that we have explained that this is necessary to approach the data to normal distributions, in their opinion, data should have been better assessed with a non-parametric method (Kruskal-Wallis) instead as such artificial transformation that may generate some false differences.

Response to comment 1: We agree that a non-parametric method may be more powerful in situations where the distributional assumptions of a parametric method are violated. In our analysis, however, it would not be appropriate to apply simple non-parametric tests for group comparisons (like Kruskal-Wallis), because we have a grouped data variable with several measurements (different brain regions) for each mouse (i.e. repeated variable). That is why we used linear mixed-effects models that account for repeated measurements within mice (and therefore account for the involved heterogeneity between observations). In this modelling framework it is common practice to apply a transformation (specifically a log-transformation) to the outcome variable before model fitting [1], as we did here.

Comment 2: The reviewer commented on the fact that while the use of ligand-binding autoradiography is correct, it does not provide any information on receptor functionality. The reviewer recommends the use of agonist-stimulated GTPgS binding autoradiography or procedures to measure receptor intracellular signalling for the measurement of receptor functionality

Response: We thank the reviewer for their valuable observation. We agree CB1 receptor autoradiography provides little information regarding the coupling capacity of these receptors to intracellular signalling molecules, and as such future studies are warranted focusing on the consequences of aging on CB1 function and signalling with the use of a combination of several techniques, including CB1 agonist [35S]GTPγS binding, intracellular ++Ca imaging and electrophysiology. The authors are indeed planning such studies as a follow up of this manuscript but currently these studies are beyond the scope of the current manuscript. Nonetheless, we believe that both the immunofluorescence and receptor autoradiography approaches offered valuable and complementary information on the impact of aging on cerebral CB1 receptor density, and, combined with measurement of endocannabinoid ligands, helped to reveal major changes in the eCB system specifically during middle-age. Receptor autoradiography is one of the very few techniques, that, unlike immunohistochemistry and immunofluorescence, enables the visualisation and distribution of “functional” sites of receptors capable of binding to its innate ligand. In addition to its specificity to receptors, which cannot always be guaranteed by immunological techniques, receptor autoradiographic binding offers precise numerical determination of receptor density in fmol/mg tissue equivalent based on a standard curve, which is not possible with immunohistochemical/immunofluorescence approaches. As such, we believe that quantitative receptor autoradiographic approach offered complementary support to the immunofluorescence assays described in this study.

We have amended the manuscript to avoid any ambiguity over the use of quantitative receptor autoradiographic binding as a proxy of functional receptor activation and proposed in the discussion that future work should focus on the consequences of aging on CB1 function and signalling with the use of a combination of several techniques including CB1 agonist [35S]GTPγS binding, intracellular ++Ca imaging and electrophysiology.

Comment 3: The reviewer expressed the opinion that autoradiography may not have the necessary anatomical resolution to detect specific signal in so small brain areas, as some of the small nuclei included in Table 1.  As such the reviewer suggests caution with those specific data, as in some cases they could represent a mere artefact.

While we accept that a minority of the brain regions analysed such as the DG are anatomically small and the anatomical boundaries between CA1, CA2 and CA3 may not be as clear cut as other brain regions might be, we are confident that the spatial anatomical resolution obtained in this autoradiographic study is adequate to accurately quantify receptor density of these regions. The rest of the brain regions analysed are substantially large and with well-defined boundaries and as such have we have no concern over the accuracy and reliability of that data.

High sensitivity and special anatomical resolution are major advantages of autoradiography.  For this reason, it has been extensively utilised by the scientific community (including the authors of this manuscript) for the visualisation of even complex differentiable receptor binding patterns in small nuclei. The choice of the radionuclid and radioligand, the quality of the tissue structure, section thinness, the selection of photographic emulsion are all major factors that can influence image resolution, and hence we designed the autoradiography study to ensure optimal resolution and at the same time limit the presence of potential artefacts.

More specifically:

  1. a) We chose to use tritium [3H] as our radionuclid of choice instead of high energy radioisotope [125I], which would have had better sensitivity. Tritium emits low energy β-particles, which travels only short distances in tissue resulting in high spatial resolution and, subsequently, enables greater distinction between anatomical structures compared to [125I]. On the other hand, radioactive decay of [125I] has a greater tissue penetration, resulting in lower image resolution because ligands with a greater distance to the detection medium also contribute to image formation.         
  2. b) We used tritium sensitive Kodak MR film instead of phosphor imaging plates to obtain our images, despite the high apposition time. The reasoning behind this is the frequent appearance of artefacts and residual 'ghost images' after repeated usage of the plates.
  3. c) We cut the brain into thin sections (20 mm) and ensured tissue preservation and quality (use of optimal temperature in cryostat, freezing section at -20 and drying them following sectioning). The excellent tissue quality and the accurate skilful sectioning of the brains not only contributed to the high resolution obtained but limited the presence of any potential artefacts.
  4. d) Of note, quantitative analysis of CB1 binding in brain regions was carried out blindly and independently by two skilled researchers who had significant mouse brain neuroanatomy knowledge and experience in using the MCID image analyser. The measurements obtained by the two individuals was very similar. The MCID analyser is a purpose-built system with high resolution display (typically 1280 x 1024 pixels x 24 bits). Hence the use of this software together with the use of a large monitor assisted with the accurate and precise determination of binding in even the smallest nuclei analysed and the detection of any potential artefacts. In addition, the widespread and distinct pattern of regional variation of CB1 binding helped with the accurate identification of even the smallest regions. Of note, brain regions were always identified at the level of predetermined brain bregma coordinates by reference to the atlas of Franklin and Paxinos.

Assuming that the reviewer is referring to CA1, CA2, CA3 and DG as the potential small regions of concern, we would like to point out that the age dependent significant changes observed in these regions are entirely consistent and in line with those observed in the whole hippocampus, thus giving us more confidence of the reliability of the data. The presence of potential artefacts which would naturally lead to misleading data is routinely scrutinised in our laboratory. However, no artifacts due to [3H] quenching or other problems were detected in this study.

Finally, we would like to notify the reviewers of an error on the name of a brain region associated with an abbreviation. LGPC abbreviation currently found in the legend of table 1 is not “Lateral geniculate nucleus, parvocellular part” but “Lateral Globus Pallidus”. This has been amended in the manuscript

Comment 4: One important CNS structure of high interest in this type of studies is the cerebellum. The reviewer is requesting the potential addition of any data of endocannabinoid levels, DAGL and CB1 immunostaining and CB1 binding autoradiography in this structure

Response to comment 4: Cerebellum is indeed one of the major brain areas where the CB1 receptor expression is very high. One of the main aims of our study was to understand the relation between brain ageing and cannabinoid system activity. Thus, we focused on brain areas which are affected by physiological ageing such as cortex, hippocampus, putamen, thalamus accumbens [2] and other fore- and middle-brain regions. As other brain regions are largely preserved during ageing we stopped slicing and tissue collection before we reached it. We now added the following text to the discussion: Importantly, age affects brain in an area-specific manner: reduction in white-matter and cell loss is reported in the cortex, hippocampus, putamen, thalamus or striatum, whereas other brain regions remain largely preserved during physiological ageing [2]. Thus, cerebellum, which is a major brain area with an exceptionally high CB1 receptor expression [3] was not investigated in the present study. 

Comment 5. Lines 44-48: Authors should differentiate the enzymes involved in the generation of endocannabinoids to those involved in their degradation. 

Response: We thank the reviewer for their comment. These lines have been amended accordingly

Comment 6. Line 106: The reviewer argues that in terms of the metabolism of endocannabinoids, it is not correct to say that arachidonic acid is the precursor of 2-AG and AEA, as both are synthesized from more complex lipids, of course containing arachidonic acid. 

Response: We thank the reviewer for their comment. The line has been amended accordingly

Comment 6. Line 258: AEA, not EAE.

Response: this has now been changed.

References

  1. Bilkei-Gorzo, A.; Rácz, I.; Michel, K.; Darvas, M.; Maldonado, R.; Zimmer, A. A Common Genetic Predisposition to Stress Sensitivity and Stress-Induced Nicotine Craving. Biol. Psychiatry 2008, 63, 164–171, doi:10.1016/j.biopsych.2007.02.010.
  2. Bilkei-Gorzo, A. The Endocannabinoid System in Normal and Pathological Brain Ageing. Philos. Trans. R. Soc. B Biol. Sci. 2012, 367, 3326–3341, doi:10.1098/rstb.2011.0388.
  3. Marcaggi, P. Cerebellar Endocannabinoids: Retrograde Signaling from Purkinje Cells. Cerebellum 2015, 14, 341–353, doi:10.1007/s12311-014-0629-5.

Round 2

Reviewer 2 Report

Authors have considered some of the comments suggested on the first version, then modifying the text accordingly, but they ignored some others. The reasons for this last decision are not convincing to me in two cases: transformation of data to log scales, and anatomical resolution in autoradiographic analysis, but, if the editor believes that authors' reasons are correct, this will be OK to me.

Author Response

Dear Reviewer, For the statistical question we asked the opinion of 2 independent biostatistical experts and we wrote our response as they suggested. The part about autoradiography was written by Prof. Bailey, who has almost 20 years experience with this technique, which he himself teaches and uses at the St. George´s University of London.   Therefore, we are quite convinced about the validity of the data presentation and interpretation.   Joining to your enquiry we asked also the opinion of the Editor in an email to decide in these debated points